# Vitamin D Status in Japanese Adults: Relationship of Serum 25-Hydroxyvitamin D with Simultaneously Measured Dietary Vitamin D Intake and Ultraviolet Ray Exposure

**DOI:** 10.3390/nu12030743

**Published:** 2020-03-11

**Authors:** Keiko Asakura, Norihito Etoh, Haruhiko Imamura, Takehiro Michikawa, Takahiro Nakamura, Yuki Takeda, Sachie Mori, Yuji Nishiwaki

**Affiliations:** 1Department of Environmental and Occupational Health, School of Medicine, Toho University, 5-21-16, Omori-Nishi, Ota-ku, Tokyo #143-8540, Japan; haruhiko.imamura@med.toho-u.ac.jp (H.I.); takehiro.michikawa@med.toho-u.ac.jp (T.M.); takahiro.nakamura@med.toho-u.ac.jp (T.N.); md18014t@st.toho-u.jp (Y.T.); ksachie82@gmail.com (S.M.); yuuji.nishiwaki@med.toho-u.ac.jp (Y.N.); 2Department of Biomedical Engineering, School of Engineering, Tokai University, 143 Shimokasuya, Isehara, Kanagawa #259-1193, Japan; norihito@tsc.u-tokai.ac.jp

**Keywords:** vitamin D, ultraviolet ray exposure, dietary intake, quantitative relationship

## Abstract

Vitamin D insufficiency/deficiency is prevalent worldwide. We investigated the effect of vitamin D intake and ultraviolet ray (UV) exposure on serum vitamin D concentration in Japan. A total of 107 healthy adult participants were recruited from Hokkaido (43° N) and Kumamoto (33° N) prefectures. All participants undertook surveys in both summer and winter. Serum 25-hydroxyvitamin D (25(OH)D_3_) was examined, and vitamin D intake was assessed with a diet history questionnaire. UV exposure was measured with a wearable UV dosimeter. Regression analysis was performed to investigate the relationship between these factors, with covariates such as sun avoidance behavior. The prevalence of vitamin D insufficiency (serum 25(OH)D_3_; 12 ng/mL (30 nmol/L) ≤ and <20 ng/mL (50 nmol/L))/deficiency (<12 ng/mL) was 47.7% in summer and 82.2% in winter. UV exposure time was short in Kumamoto (the urban area), at 11.6 min in summer and 14.9 min in winter. In Hokkaido (the rural area), UV exposure time was 58.3 min in summer and 22.5 min in winter. Vitamin D intake was significantly associated with serum 25(OH)D_3_, and a 1 μg/1000kcal increase in intake was necessary to increase 25(OH)D_3_ by 0.88 ng/mL in summer and by 1.7 ng/mL in winter. UV exposure time was significantly associated with serum 25(OH)D_3_ in summer, and a 10 min increase in UV exposure time was necessary to increase 25(OH)D_3_ by 0.47 ng/mL. Although consideration of personal occupation and lifestyle is necessary, most Japanese may need to increase both vitamin D intake and UV exposure.

## 1. Introduction

The fat-soluble vitamin D has two sources in humans—dietary intake and synthesis in the skin following exposure to ultraviolet B (UVB) radiation with wavelengths from 290 to 320 nm [1]. Vitamin D has important functions in bone health [1], and several recent studies have investigated other positive health effects, including the prevention of cancer [2] and cardiovascular disease [3]. Findings for these non-skeletal outcomes remain inconclusive, however, and current dietary recommendations for vitamin D have been designed to maintain serum 25-hydroxyvitamin D (25(OH)D) at concentrations sufficient to maintain healthy bones [1].

Despite the importance of vitamin D, deficiency (serum 25(OH)D; <12 ng/mL (30 nmol/L)) and insufficiency (12 ng/mL ≤ and <20 ng/mL (50 nmol/L)) have been reported worldwide [4,5].They are also prevalent in Japan: one study reported the proportion of pregnant Japanese women with low (<20 ng/mL) serum 25(OH)D was 89.8% in April (Spring) and 47.8% in October (Autumn) [6], while another group reported a similar rate of 41.6% in April and May among Japanese workers, mostly men [7].

While several countries have established dietary reference values for vitamin D intake to avoid deficiency [8], these efforts have been challenging. The lack of UV measurement tools has hampered efforts to estimate the effect of UV exposure on vitamin D status in the body, and reference values in several northern countries were established assuming that UVB exposure was minimal. Examples include the North American Institute of Medicine (IOM) [1] and Nordic Nutrition Recommendations working group [9], which recommends values of 10–20 μg/day. In Japan, a reference value called “adequate intake (AI)” has been established [10], assuming that a proportion of vitamin D comes from cutaneous synthesis. This value is set when there is insufficient evidence to establish firmer indexes, such as estimated average requirements and recommended dietary allowance. AI for Japanese adults of 5.5 μg/day is much lower than that of other countries [10]. The Japanese Dietary Reference Intakes (DRI) indicate the need for more information, particularly about UV exposure [10].

A recent device, the UV Dosimeter Badge, quantitatively measures personal UV exposure. The utility and reliability of this device have been confirmed [11]. Here, we used the UV Dosimeter Badge to simultaneously measure serum 25(OH)D, vitamin D intake and UV exposure, and estimate the respective contributions of vitamin D intake and UV exposure to serum vitamin D concentration in Japan, a middle-latitude country.

## 2. Materials and Methods 

### 2.1. Study Design and Participants

We recruited apparently healthy adult volunteers aged 20 to 69 years from two areas of Japan, in cooperation with local governments. The northern area was Shakotan Town, Hokkaido prefecture, located at 43 degrees north. Shakotan is located in a rural area with approximately 2,000 residents, and fishery and tourism are the major industries. In 2018, max/min temperature in summer and winter was 34.0/4.4 °C and 13.2/−16.9 °C, respectively. It frequently snows in the town in winter. The southern area was Kumamoto City, in Kumamoto prefecture at 33 degrees north, an urban area with approximately 740,000 residents and tertiary industry as the major activity. In the same year, max/min temperature in summer and winter was 38.1/15.7 °C and 24.0/−5.2 °C, respectively. The monthly means of max/min UV index in Hokkaido and Kumamoto in 2018 were 0.7 (December)/5.4 (July) and 1.9 (January and December)/8.4 (August), respectively. The target number of participants from each area was 60 (120 in total), with 12 participants (six men and six women) recruited from each of five 10-year age groups (20–29, 30–39, 40–49, 50–59, and 60–69 years) to adjust for the effects of age and sex. This number was decided mostly on the basis of feasibility, considering the burden of the surveys, population and age structure of the survey area, and limited number of UV Dosimeter Badges. Inclusion criteria were (1) age 20–69 years, (2) living or working in one of the study areas, (3) ability to answer questionnaires written in Japanese, and (4) skin color of East Asians. We excluded pregnant and lactating women to avoid the influence of temporal change in dietary habits. In total, 120 subjects participated.

### 2.2. Measurement Schedule

The survey schedule and items are shown in Figure 1. The study was performed in accordance with the Declaration of Helsinki and was approved by the Ethics Committee of the Faculty of Medicine, Toho University (approval no. A18035_A17031, July 10, 2017). In each area, the surveys were performed twice, at the end of summer and at the end of winter to catch the highest and lowest serum vitamin D values (Figure 1, (1)). The Japan Meteorological Agency defines summer as being from June to August, and winter from December to February. Regarding people living in Japan, there is no previous work showing in which month vitamin D synthesis in the skin starts to affect serum 25(OH)D concentration.

Participants were asked to participate in both surveys by local government officials in health-related departments orally, by telephone, or by email. After obtaining written informed consent by the researchers, a UV measurement device, two questionnaires, and an activity diary were distributed to the participants on the first survey day. They wore the device for 10–14 days, and kept the activity diary with a focus on factors relating to UV exposure (clothing, sunscreen use, duration of outdoor activity). They also completed the dietary assessment and lifestyle questionnaires during this period. After the UV measurement period, the participants visited the survey site to return the UV measurement device and submit the questionnaires. At this time, blood specimens were collected and body height/weight were measured.

### 2.3. Ultraviolet Ray Exposure Measurement

The UV Dosimeter Badge (Scienterra Limited, Otago, New Zealand), a round, button-like electric device of 36 mm diameter, 12 mm thick, and 26 grams, was used to measure exposure to UV of shorter wavelengths (UVB) [11]. Each participant wore it on the upper arm of their nondominant hand with an arm band. The badge measures UVB power (μW/cm^2^) irradiated on its sensor every two seconds from sunrise to sunset. Stored data is offloaded to a computer using a docking cradle, which communicates with the badge through a high-speed optical interface. For this study, UV exposure time (minutes/day) was calculated as cumulative time (minutes) in which we observed UVB irradiation on the badge across all measurement days divided by the number of measurement days. Crude UV exposure energy (J/cm^2^/day) was calculated as the time-integrated value of UVB-irradiated power at each moment during the day. For this analysis, UV exposure energy was adjusted by body surface area, irradiated body site, clothing and sunscreen use per day. Adjusted UV exposure energy (J/day) for each participant was calculated (Figure 2). Total body surface area (cm^2^) was calculated according to Kurazumi’s formula [12], and body proportion factor for each body site was set as reported by Lund and Browder [13]. UV power adjustment by irradiated body site [14] and by hat [15] was performed based on previous studies. If sunscreen was used, UV power was multiplied by 0.9, albeit there is scarce evidence that sunscreen decreases serum 25(OH)D concentration [16]. We used 0.9 in consideration of the likely use of sunscreen with a high sun protection factor (SPF). Information about the weather during the survey period was not utilized, because this information was already reflected in the measured UV exposure energy.

### 2.4. Dietary Assessment

Dietary intake of vitamin D was assessed using a comprehensive self-administered diet history questionnaire (DHQ), a 22-page semi-quantitative questionnaire that assessed habitual dietary intake during the month preceding implementation of the survey. Details and results of validation studies of the DHQ have been reported elsewhere [17,18,19]. For instance, Shiraishi et al reported the validity of the DHQ for estimating vitamin D intake in pregnant Japanese women. Pearson’s correlation coefficients between dietary vitamin D intakes and serum 25(OH)D concentration were 0.304 (*p* = 0.001) in all subjects and 0.371 (*p* = 0.001) in those without nausea in winter, the season in which serum 25(OH)D concentrations are less likely to be affected by sunlight. Submitted DHQs were checked by trained research staff immediately after collection, who followed up with the respondent directly over unclear points. Estimates of daily intake of foods, energy, and nutrients were calculated based on the Standard Tables of Food Composition in Japan [20]. Before analysis, vitamin D intake was energy-adjusted by the density method, wherein daily vitamin D intake was calculated as the amount (μg) per 1000 kcal of daily energy intake [21]. In Japan, supplement use of vitamin D is uncommon, given that only 10%–30% of people are estimated to use supplements of any type [22,23,24]. Vitamin D fortification of foods is rare.

### 2.5. Blood Tests

Serum 25(OH)D was measured by liquid chromatography–tandem mass spectrometry (LC-MS/MS) at LSI Medience Corporation (Tokyo, Japan). The company used ‘6PLUS1 Multilevel Serum Calibrator Set 25-OH-Vitamin D3/D2’ and ‘MassCheck 25-OH-Vitamin D3/D2 Serum Control, Bi-Level (I+II)’ for calibration and validation of the measurements in accordance with the company’s application notes. These materials are supplied by Chromsystems (Gräfelfing, Germany) and are verified by National Institute of Standards and Technology’s standards. Interassay CV for 25(OH)D_3_ measurement was 4.9%. Serum concentrations of 25(OH)D_2_ and 25(OH)D_3_ were measured, but only 25(OH)D_3_ values were used as index because all values of 25(OH)D_2_ were below the lower limit of measurement. In accordance with the DRI for calcium and vitamin D in the US and Canada [1], a serum concentration of 25(OH)D_3_ of ≥ 20 ng/mL (50 nmol/L, 1ng/mL = 2.5nmol/L) was considered to indicate sufficiency of vitamin D in the body. Similarly, serum 25(OH)D_3_ <12 ng/mL (30 nmol/L) was defined as vitamin D deficiency, while a value between these cutoffs (12 ng/mL ≤ 25(OH)D_3_ <20 ng/mL) was considered the range of vitamin D insufficiency. In addition to 25(OH)D, the intact parathyroid hormone (PTH) was measured by electrochemiluminescence immunoassay (ECLIA) as an index of serum calcium status and bone metabolism.

### 2.6. Questionnaire and Other Measurements

Information on participants’ backgrounds and lifestyle factors, particularly those related to UV exposure, was obtained by a self-administered questionnaire. Skin type was based on the Fitzpatrick phototyping scale [25], and physical activity in summer was based on the International Physical Activity Questionnaire (IPAQ) [26,27].

Body height and weight were measured to the nearest 0.1 cm and 0.1 kg, respectively, in light clothing and no shoes. BMI was calculated as body weight in kilograms divided by the square of body height in meters.

### 2.7. Statistical Analysis

A total of 107 participants who completed all surveys in summer and winter were included. Characteristics were summarized by study area and sex. The relationship of serum 25(OH)D_3_ with dietary vitamin D intake and UV exposure was then depicted in scatter plots, with data from Hokkaido and Kumamoto shown in the same plot by season. Finally, we examined the relationship by linear regression analysis. In the models, serum 25(OH)D_3_ was a dependent variable, with dietary vitamin D intake and UV exposure (either time (model 1_time_, 2_time_) or energy (model 1_energy_, 2_energy_)) simultaneously included as independent variables. Sex (reference: men) and study area (reference: Hokkaido) were adjusted in models 1_time_ and 1_energy_. Age was excluded from models 1_time_ and 1_energy_ to avoid over-adjustment, because it was positively and significantly associated with vitamin D intake and UV exposure. Other covariates were selected from factors possibly related to serum vitamin D concentration and/or cutaneous vitamin D synthesis [1,28]. In model 2_time_, age, smoking, skin type, sunscreen use (nonuser/user), sunshade clothing/tools use (nonuser/user (any or all of hat, gloves, and parasol)), and physical activity (summer only) were included, in addition to the variables included in the model 1_time_. In model 2_energy_, age, smoking, skin type, and physical activity (summer only) were included, in addition to those included in the model 2_time_, because UV exposure energy had already been adjusted by clothing and sunscreen use. All analyses were performed with SAS version 9.4 (SAS Institute, Cary, NC, USA). Statistical tests were two-sided, and P values of <0.05 were considered statistically significant.

## 3. Results

Participants’ characteristics are summarized in Table 1. All participants completed the surveys both in summer and winter. Average serum 25(OH)D_3_ among all subjects (*n* = 107) was 21.1 ng/mL in summer and 14.6 ng/mL in winter, and was marginally or significantly higher in Hokkaido (t test, *p* = 0.07 in both summer and winter) and in men (*p*<0.001 in summer, *p* = 0.08 in winter). The prevalence of vitamin D insufficiency/deficiency was 47.7% in summer and 82.2% in winter. The greatest contributor to total vitamin D intake was fish, which accounted for 70.9% of total intake both in summer and winter. The second contributor was eggs (12.0% in summer and 12.3% in winter), followed in order by confectionaries (4.4% and 4.7%) and mushrooms (4.0% and 4.5%, respectively).

The relationships between serum 25(OH)D_3_ and dietary vitamin D intake, and between serum 25(OH)D_3_ and UV exposure time/energy in summer and winter are shown in scatter plots in Figure 3. All regression lines rose to the right. Distributions of UV exposure, particularly UV exposure energy, were narrower in winter.

Linear regression analyses are shown in Table 2 and Table 3. In both summer and winter, vitamin D intake, UV exposure time and sex were significantly associated with serum 25(OH)D_3_ in Model 1_time_ (Table 2). In contrast, area was not associated with serum 25(OH)D_3_. In Model 2_time_, no added covariate except age was associated with serum 25(OH)D_3_. UV exposure time was not significantly associated with serum 25(OH)D_3_ in winter in Model 2_time_, but became marginal and the regression coefficient changed to 0.057 (95% CI, -0.003 to 0.117) when age was eliminated. In Model 2_time_, a 1 μg/1000kcal increase in vitamin D intake was necessary to increase serum 25(OH)D_3_ by 0.88 ng/mL in summer and 1.7 ng/mL in winter. Similarly, a 10 min increase in UV exposure time was necessary to increase serum 25(OH)D_3_ by 0.47 ng/mL in summer and 0.41 ng/mL in winter. Results were similar when UV exposure energy was used in the models (Table 3). The difference in results was the non-significant relationship between serum 25(OH)D_3_ and UV exposure energy in Model 1_energy_ in winter. Results were similar when the coefficient for sunscreen use (0.9) was not used to adjust for UV exposure energy.

## 4. Discussion

This study is the first to measure UV exposure (both time and energy) and vitamin D intake, and to investigate the relationship between them simultaneously; this study is also the first to perform standardized measurements in both summer and winter in both northern and southern Japan. In particular, this study is the first to compare the influence of time of UV exposure with energy of exposure and serum 25(OH)D levels, with both being objectively measured via electronic dosimeters. Results showed that the prevalence of vitamin D insufficiency/deficiency was high, and was rather higher in (southern) Kumamoto than (northern) Hokkaido, suggesting that the effect of lifestyle factors, such as dietary habits and length of outdoor activity, had greater influence on vitamin D status in the body than this difference in latitude.

Vitamin D intake did not considerably differ between the two areas, at around 3 μg/1000kcal (6 μg/2000kcal: approximate of daily intake). This intake is the same as the AI (5.5 μg/day) in the Japanese DRI [10]. Given the high prevalence of vitamin D insufficiency in Japan, however, these findings indicate that most Japanese should consume more vitamin D, even in summer. A new version of the Japanese DRI (version 2020) is now being adopted; in this, the AI is raised from 5.5 to 8.5 μg/day for adults. Close observation is required to determine the effect of this change.

Perhaps surprisingly, UV exposure time was longer and energy was stronger among participants in Hokkaido than Kumamoto, particularly in summer (Table 1). Most participants in Kumamoto were office workers. Their mean UV exposure time of 11.6 min in summer and 14.9 min in winter suggests that exposure primarily occurred when out of the office, such as in short walks during commuting, going out for lunch, or shopping. Since 71.0% of employed Japanese work in tertiary industries such as retailing, food service and financial businesses [29], the lifestyle observed in Kumamoto is likely representative of Japanese living in cities. Longer outdoor activity would be preferable for people with short UV exposure time. While an appropriate length of UV exposure requires further discussion, the average UV exposure time we observed in Kumamoto may act as a reference. In addition, this study is the first to estimate and adjust UV exposure energy by exposed body surface area, irradiated body site, and use of sunscreen/clothing. In winter, UV exposure energy was not significantly associated with serum 25(OH)D_3_. This non-significant relationship may have been due to the low energy itself and its narrow distribution, and the importance of vitamin D intake would appear to be greater in winter. In addition, although we asked the participants to wear the dosimeter over the top of their clothing, it was more difficult to measure and adjust UV exposure power in winter, because most participants wore jackets and cold protectors.

Female participants and those of a younger age were significantly associated with lower serum 25(OH)D_3_. Women tended to avoid tanning, using sunscreen and sunshade clothing, and this might have caused lower UV exposure energy in women, particularly in summer. Regarding age, given findings that increases in serum 25(OH)D by vitamin D intake do not vary with age up to at least 80 years [30,31], whereas the capacity to produce vitamin D_3_ in skin decreases with aging [32], it is difficult to assume that serum 25(OH)D_3_ physiologically increases with age. The higher serum 25(OH)D_3_ in older participants appears to be due to external factors, namely a higher intake of fish as a major vitamin D source and longer UV exposure (Appendix A). Fish intake in the participants in their twenties was 19.8g/1000kcal in summer and 25.1 g/1000kcal in winter, versus respective values in those in their sixties of 33.8 and 29.6 g/1000kcal. Further, UV exposure time was 16.1 min in summer and 15.0 min in winter in those in their twenties, versus 68.4 and 26.4 min in those in their sixties, respectively. Since half of the participants in their sixties were housewives or post-retirement, they may have had more time to perform outside activities.

Sunscreen and sunshade clothing/tools use also showed no association with 25(OH)D_3_ in the multivariate models (Table 2). Most sunscreen users were women, and sunshade tool use substantially differed between men and women. As sex was significantly associated with serum 25(OH)D_3_, we were unable to rule out sex as a possible confounding factor. In addition, physical activity was not associated with serum 25(OH)D_3_ in summer. Some previous studies did report such an association [33,34], but this may likely have been a surrogate marker of UV exposure. There are some past studies suggesting a relationship between body mass index (BMI) and serum vitamin D concentration [35]. However, we did not include BMI in the statistical models, because we observed no relationship between them.

As shown in Figure 3 and Table 2 and Table 3, we assumed that the relationship of serum 25(OH)D_3_ with vitamin D intake and UV exposure was linear. Based on this assumption, the regression coefficients shown in Table 2 and Table 3 indicated the effect of the amount of dietary vitamin D intake and UV exposure to serum 25(OH)D_3_. However, some studies have suggested that these relationships are not linear [1]. Aloia et al. reported that the response of serum 25(OH)D to 1 μg of vitamin D_3_ intake was inversely dependent on the basal 25(OH)D concentration [36]. Similarly, Olds et al. suggested that the relationship between sun exposure and serum 25(OH)D level was curvilinear [37]. Interestingly, excess previtamin D_3_ and vitamin D_3_ is destroyed by sunlight [38]. Therefore, the effect of dietary vitamin D intake and UV exposure on people with vitamin D insufficiency may be larger than the coefficients in this study. This homeostatic mechanism might also explain the smaller difference in serum 25(OH)D_3_ concentration in summer between Hokkaido and Kumamoto compared with the larger difference in UV exposure.

A few studies have investigated internal vitamin D status with quantitative measurement of UV exposure. Callegari et al. reported that personal sun exposure measured by the same Dosimeter Badge as used here was significantly associated with serum 25(OH)D_3_, but did not report the length of UV exposure and did not include vitamin D intake in their regression models [39]. Scragg et al. also used this dosimeter, but also did not investigate vitamin D intake [40]. O’Sullivan et al. estimated the daily ambient UVB dose for each participant in their study by using irradiated UV energy at various geographical points in Ireland and the residential address of the participants, and reported a relationship between it and serum 25(OH)D concentration [41]. Their estimation method differed from ours, which directly measured UV energy on the participants’ arms.

Several limitations of this study should be mentioned. First, the participant number was small and sampling was not random. Accordingly, the participants were likely to have been more health-conscious than the general population. Since the burden of the survey was heavy, random sampling was not feasible. Although the target number for recruitment was decided mostly based on feasibility, it seemed that the number of participants was appropriate to detect differences of behavior affecting serum vitamin D status, because the difference was large. For example, to detect whether differences in UV exposure time between participants with and without sufficient serum vitamin D were significant using a two-sample t test, a total of 90 subjects in summer and 81 subjects in winter were required, with an alpha equal to 0.05 and statistical power of 0.80 (calculated by proc power procedure of SAS). Second, the occupation of the participants substantially differed between Hokkaido and Kumamoto. This caused a difference in UV exposure, and may have masked the effect of latitude. Nevertheless, our results are consistent with other studies showing that latitude alone does not always predict average serum vitamin D level in a population [42]. Third, since we adopted self-reported dietary assessment, misreporting was possible. However, the questionnaire was validated and responses were checked carefully, likely minimizing measurement errors. Fourth, this study is the first to adjust UV exposure by the area of the body exposed and sun avoidance behavior, and the coefficients used in the adjustment require further consideration. Finally, as described above, the relationship of serum 25(OH)D_3_ with UV exposure and vitamin D intake may not be linear. This study is the first to show these regression coefficients, i.e., effect sizes for these relationships, but they should be interpreted with care.

## 5. Conclusions

Vitamin D insufficiency/deficiency was prevalent in a Japanese population, even in a mid-latitude area. Both vitamin D intake and UV exposure affected vitamin D status in the body. Although consideration for personal occupation and lifestyle is necessary, most Japanese may need to increase both vitamin D intake and UV exposure.

## Figures and Tables

**Figure 1 nutrients-12-00743-f001:**
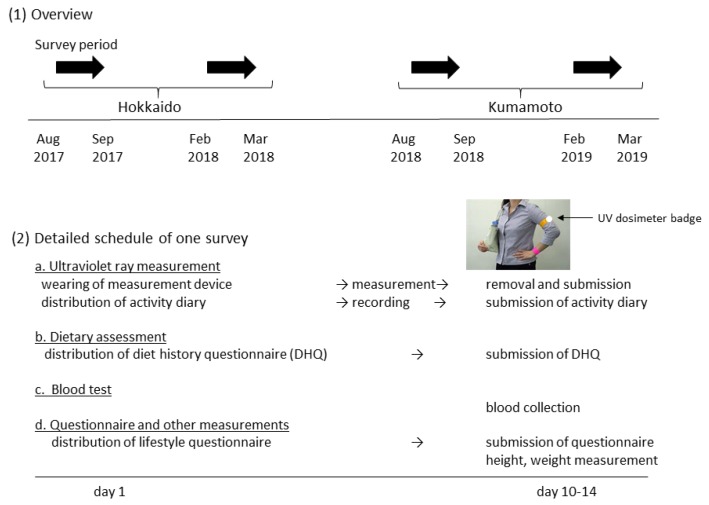
Survey schedule.

**Figure 2 nutrients-12-00743-f002:**
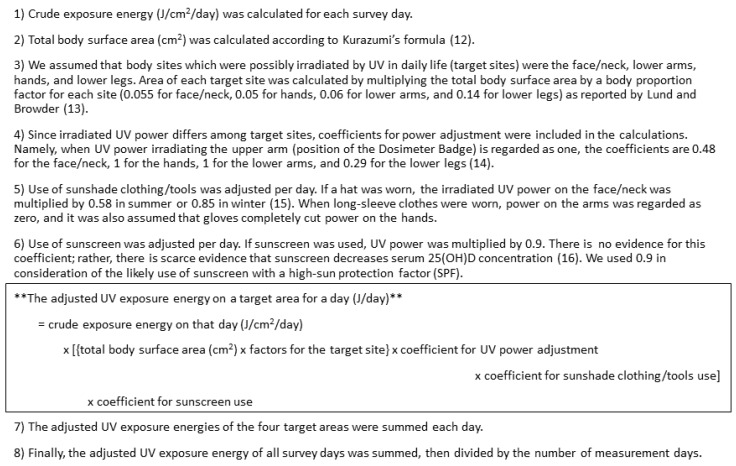
Calculation of adjusted UV exposure energy (J/day). Reference numbers correspond to those in the text.

**Figure 3 nutrients-12-00743-f003:**
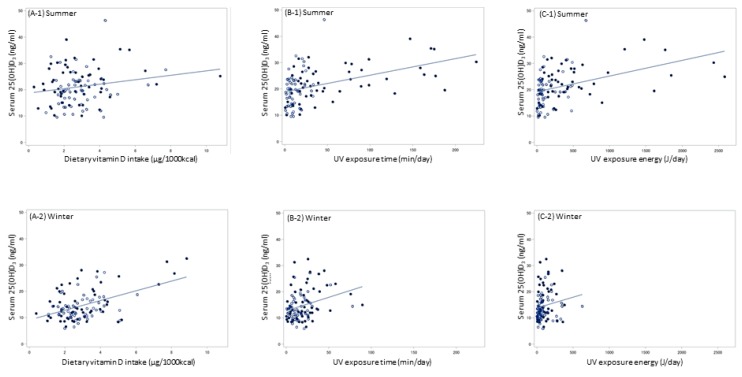
Relationship of serum 25(OH)D_3_ with dietary vitamin D intake and UV exposure; (**A**) dietary vitamin D intake (μg/1000kcal) and serum 25(OH)D_3_ concentration (ng/mL); (**B**) UV exposure time (min/day) and serum 25(OH)D_3_ concentration (ng/mL); (**C**) UV exposure energy (J/day) and serum 25(OH)D_3_ concentration (ng/mL); Symbols: ● Hokkaido (northern Japan), ○ Kumamoto (southern Japan).

**Table 1 nutrients-12-00743-t001:** Characteristics of participants (total participants: *n* = 107).

Variable	Season^a^	*n* (%) or Mean, SD							
		Study Area				Sex			
		Hokkaido (lat.43° N, *n* = 54)		Kumamoto (lat.33° N, *n* = 53)		Men (*n* = 53)		Women (*n* = 54)	
<Background>									
Age 20s	S, W	11	(20.4)	9	(17.0)	9	(17.0)	11	(20.4)
30s	S, W	9	(16.7)	11	(20.8)	11	(20.8)	9	(16.7)
40s	S, W	12	(22.2)	12	(22.6)	12	(22.6)	12	(22.2)
50s	S, W	12	(22.2)	11	(20.8)	11	(20.8)	12	(22.2)
60s	S, W	10	(18.5)	10	(18.9)	10	(18.9)	10	(18.5)
Height (cm)	S	162.8	7.5	165.0	8.0	169.8	5.6	158.1	4.6
Weight (kg)	S	61.9	11.9	63.0	15.2	69.3	13.2	55.7	10.3
BMI (kg/m^2^)	S	23.2	3.8	23.0	4.2	24.0	3.9	22.3	3.9
Skin type^b^									
II	S	14	(25.9)	18	(34.0)	11	(20.8)	21	(38.9)
III	S	27	(50.0)	25	(47.2)	28	(52.8)	24	(44.4)
IV	S	13	(24.1)	10	(18.9)	14	(26.4)	9	(16.7)
<Blood test>									
25(OH)D_3_ (ng/mL)	S	22.3	6.7	20.0	6.7	24.1	6.8	18.3	5.5
	W	15.6	6.6	13.7	4.3	15.6	5.4	13.7	5.7
Vitamin D status									
sufficient (20≤25(OH)D_3_)	S	32	(59.3)	24	(45.3)	38	(71.7)	18	(33.3)
insufficient (12≤25(OH)D_3_<20)	S	20	(37.0)	23	(43.4)	14	(26.4)	29	(53.7)
deficient (25(OH)D_3_<12)	S	2	(3.7)	6	(11.3)	1	(1.9)	7	(13.0)
sufficient (20≤25(OH)D_3_)	W	15	(27.8)	4	(7.6)	12	(22.6)	7	(13.0)
insufficient (12≤25(OH)D_3_<20)	W	21	(38.9)	27	(50.9)	26	(49.1)	22	(40.7)
deficient (25(OH)D_3_<12)	W	18	(33.3)	22	(41.5)	15	(28.3)	25	(46.3)
PTH intact (pg/mL)	S	45.4	15.4	52.5	19.0	47.4	15.1	50.5	19.7
	W	48.3	16.7	55.8	18.7	51.2	18.1	52.8	18.1
<Dietary intake>									
Energy (kcal)	S	1804	699	1954	537	2071	559	1690	635
	W	1776	497	1845	471	1968	449	1656	469
Vitamin D (μg/1000kcal)	S	2.8	1.8	2.9	1.3	2.6	1.5	3.1	1.6
	W	3.1	1.8	2.9	1.0	2.7	1.3	3.2	1.5
Fish (g/1000kcal)	S	28.5	21.8	26.3	12.4	25.7	16.9	29.0	18.5
	W	30.2	20.1	25.6	10.7	26.1	14.2	29.7	18.0
<Ultraviolet ray exposure>									
Exposure time (min/day)	S	58.3	58.7	11.6	9.7	37.8	54.6	32.6	41.2
	W	22.5	18.1	14.9	13.2	20.1	20.0	17.4	11.4
Exposure energy (J/day)	S	494	598	144	162	407	594	236	291
	W	104	93	99	118	105	118	99	93
<Lifestyle factors>									
Smoking									
Never	S	27	(50.0)	33	(62.3)	20	(37.7)	40	(74.1)
Past	S	12	(22.2)	14	(26.4)	16	(30.2)	10	(18.5)
Current	S	15	(27.8)	6	(11.3)	17	(32.1)	4	(7.4)
Occupation									
Manager	S	2	(3.7)	4	(7.5)	6	(11.3)	0	(0.0)
Professional	S	7	(13.0)	21	(39.6)	10	(18.9)	18	(33.3)
Clerical	S	18	(33.3)	21	(39.6)	24	(45.3)	15	(27.8)
Service and sales	S	7	(13.0)	1	(1.9)	4	(7.5)	4	(7.4)
Agricultural, forestry and fisheries	S	7	(13.0)	0	(0.0)	3	(5.7)	4	(7.4)
Craft and related trades	S	4	(7.4)	0	(0.0)	3	(5.7)	1	(1.9)
Others^c^	S	9	(16.7)	6	(11.3)	3	(5.7)	12	(22.2)
Physical activity (Mets x minutes)	S	496	741	164	153	368	654	296	455
Sunscreen use^d^									
Face	S	13	(24.1)	22	(41.5)	1	(1.9)	34	(63.0)
	W	11	(20.4)	17	(32.1)	0	(0.0)	28	(51.9)
Neck	S	7	(13.0)	14	(26.4)	1	(1.9)	20	(37.0)
	W	3	(5.6)	3	(5.7)	0	(0.0)	6	(11.1)
Arm	S	6	(11.1)	12	(22.6)	1	(1.9)	17	(31.5)
	W	0	(0.0)	0	(0.0)	0	(0.0)	0	(0.0)
Hand	S	6	(11.1)	10	(18.9)	1	(1.9)	15	(27.8)
	W	2	(3.7)	1	(1.9)	0	(0.0)	3	(5.6)
Any^e^	S	13	(24.1)	22	(41.5)	1	(1.9)	34	(63.0)
	W	11	(20.4)	17	(32.1)	0	(0.0)	28	(51.9)
Sunshade clothing/tools use^d^									
Hat	S	18	(33.3)	7	(13.2)	8	(15.1)	17	(31.5)
	W	27	(50.0)	5	(9.4)	18	(34.0)	14	(25.9)
Parasol	S	6	(11.1)	22	(41.5)	3	(5.7)	25	(46.3)
	W	2	(3.7)	0	(0.0)	1	(1.9)	1	(1.9)
Long sleeves	S	13	(24.1)	10	(18.9)	9	(17.0)	14	(25.9)
	W	52	(96.3)	53	(100.0)	52	(98.1)	53	(98.2)
Gloves	S	11	(20.4)	12	(22.7)	5	(9.4)	18	(33.3)
	W	35	(64.9)	14	(26.4)	25	(47.2)	24	(44.5)
Combination^f^	S	24	(44.4)	24	(45.3)	12	(22.6)	36	(66.7)
	W	37	(68.5)	17	(32.1)	28	(52.8)	26	(48.2)

Body mass index (BMI); latitude (lat); parathyroid hormone (PTH); 25-hydroxyvitamin D_3_ (25(OH)D_3_); ^a^ Season in which the survey was performed. Summer (S), winter (W); ^b^ Fitzpatrick skin type was self-reported. The classification was made by erythema and tanning reactions to first exposure to sunlight in summer. II: usually burn, tan less than average, III: sometimes mildly burn, tan about average, IV: rarely burn, tan more than average; ^c^ The “Other” category of occupation included housewives, retirees, and so on; ^d^ Multiple answers allowed; ^e^ The “Any” category of sunscreen use includes all sunscreen users irrespective of application site; ^f^ The “Combination” category of sunshade clothing/tools use includes users who used any or all of hat, parasol, and gloves.

**Table 2 nutrients-12-00743-t002:** Relationship between dietary vitamin D intake, UV exposure time and serum 25(OH)D_3_ concentration by season among Japanese adults (*n* = 107).

Season	Variable	Unit, Category (Ref)	Model 1_time_			Model 2_time_		
			Regression Coefficient	95%CI	*p* Value	Regression Coefficient	95%CI	*p* Value
Summer	VD intake	μg/1000kcal	1.00	(0.34, 1.67)	0.004	0.88	(0.14, 1.62)	0.020
	UV exposure time	min	0.059	(0.034, 0.083)	<0.001	0.047	(0.014, 0.079)	0.005
	Sex	women (vs men)	−5.94		<0.001	−5.39		0.001
	Area	Kumamoto (vs Hokkaido)	0.34		0.77	−0.17		0.90
	Age	years				0.08		0.08
	Smoking	past smoker (vs nonsmoker)				−0.12		0.93
		current smoker (vs nonsmoker)				−0.76		0.62
	Skin type^a^	III (vs II)				−0.46		0.71
		IV (vs II)				0.25		0.87
	Sunscreen use^b^	user (vs nonuser)				−1.67		0.29
	Sunshade clothing/tools use^c^	user (vs nonuser)				0.53		0.70
	Physical activity^d^	Mets x min				0.00		0.93
Winter	VD intake	μg/1000kcal	1.88	(1.25, 2.52)	<0.001	1.70	(1.09, 2.32)	<0.001
	UV exposure time	min	0.067	(0.0097, 0.125)	0.022	0.041	(−0.016, 0.099)	0.16
	Sex	women (vs men)	−2.65		0.005	−1.90		0.10
	Area	Kumamoto (vs Hokkaido)	−1.05		0.26	−0.73		0.45
	Age	years				0.12		0.001
	Smoking	past smoker (vs nonsmoker)				−0.92		0.42
		current smoker (vs nonsmoker)				0.61		0.63
	Skin type^a^	III (vs II)				0.42		0.69
		IV (vs II)				−0.27		0.83
	Sunscreen use^b^	user (vs nonuser)				−1.27		0.33
	Sunshade clothing/tools use^c^	user (vs nonuser)				1.11		0.24

Confidence interval (CI); minutes (min); reference category (ref); vitamin D (VD); ultraviolet ray (UV); 25-hydroxyvitamin D_3_ (25(OH)D_3_); ^a^ Fitzpatrick skin type was self-reported. The classification was made by erythema and tanning reactions to first exposure to sunlight in summer. II: usually burn, tan less than average, III: sometimes mildly burn, tan about average, IV: rarely burn, tan more than average; ^b^ Nonuser means those who did not use sunscreen at all. Those who used sunscreen on any body surface were considered users; ^c^ Those who used any or all of hat, parasol, and gloves. The others were considered nonusers; ^d^ Physical activity level was estimated by the International Physical Activity Questionnaire (IPAQ) only in summer.

**Table 3 nutrients-12-00743-t003:** Relationship between dietary vitamin D intake, UV exposure energy and serum 25(OH)D_3_ concentration by season among Japanese adults (*n* = 107).

Season	Variable	Unit, Category (Ref)	Model 1_energy_			Model 2_energy_		
			Regression Coefficient	95%CI	*p* Value	Regression Coefficient	95%CI	*p* Value
Summer	VD intake	μg/1000kcal	1.10	(0.41, 1.79)	0.002	0.81	(0.083, 1.55)	0.029
	UV exposure energy^a^	J/day	0.004	(0.0020, 0.0069)	0.001	0.004	(0.001, 0.007)	0.012
	Sex	women (vs men)	−5.52		<0.001	−5.80		<0.001
	Area	Kumamoto (vs Hokkaido)	−0.85		0.46	−1.41		0.23
	Age	years				0.12		0.005
	Smoking	past smoker (vs nonsmoker)				0.02		0.99
		current smoker (vs nonsmoker)				−1.01		0.51
	Skin type^b^	III (vs II)				−0.51		0.69
		IV (vs II)				0.15		0.93
	Physical activity^c^	Mets x min				0.00		0.90
Winter	VD intake	μg/1000kcal	1.94	(1.29, 2.58)	<0.001	1.75	(1.13, 2.36)	<0.001
	UV exposure energy	J/day	0.005	(−0.004, 0.013)	0.29	0.004	(−0.005, 0.012)	0.37
	Sex	women (vs men)	−2.83		0.003	−2.66		0.007
	Area	Kumamoto (vs Hokkaido)	−1.53		0.10	−1.58		0.08
	Age	years				0.14		<0.001
	Smoking	past smoker (vs nonsmoker)				−0.72		0.53
		current smoker (vs nonsmoker)				0.61		0.63
	Skin type^b^	III (vs II)				0.59		0.57
		IV (vs II)				−0.08		0.95

Confidence interval (CI); minutes (min); reference category (ref); vitamin D (VD); ultraviolet ray (UV); 25-hydroxyvitamin D_3_ (25(OH)D_3_); ^a^ UV exposure energy adjusted by exposed body surface area, irradiated body site, as well as sunscreen and sunshade clothing/tools use; ^b^ Fitzpatrick skin type was self-reported. The classification was made by erythema and tanning reactions to first exposure to sunlight in summer; II: usually burn, tan less than average, III: sometimes mildly burn, tan about average, IV: rarely burn, tan more than average; ^c^ Physical activity level was estimated by the International Physical Activity Questionnaire (IPAQ) only in summer.

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
