# Peer review of "Vitamin D Status in Japanese Adults: Relationship of Serum 25-Hydroxyvitamin D with Simultaneously Measured Dietary Vitamin D Intake and Ultraviolet Ray Exposure"

_nutrients, 2020, doi:10.3390/nu12030743_

Round 1

Reviewer 1 Report

This a very interesting and generally well written manuscript – suggestions are detailed below

1). It may be more useful to use nmol/L rather than ng/ml which is an older term and less widely used particularly in both Europe and internationally. If still want to use ng/ml please provide the conversion factor for those not familiar with vitamin D measurements

2). Line 41 -please define what you mean by deficiency, insufficiency and sufficiency as many researchers have different criteria

3). Would be useful to provide information on when the Japanese winter and summer begins (what months) and when does UV vitamin D synthesis start and begin (from previous work or state that it is unknown if no other previous work)   

4). Lines 63-75 – no information is given on how these people were recruited, inclusion and exclusion criteria, recruitment rates, who are these people in the study -random people off the street or in a clinic/hospital?? etc. Please provide information

5). Was the vitamin D questionnaire used fully validated?

6). Lines 112-121 – what were the inter and intra assay CVS for the vitamin D, PTH and calcium biochemical analysis?

7). Lines 112-115- was an external vitamin D set of standards also used for verification and standardisation? Was an in-house kit used for the analysis or a commercial kit used e.g. Chromsystems method etc…

8). Line131 – why 107 people- was this from a power calculation (please show)

9). Is there any information on cloud or weather patterns which would in turn affect UV penetration to the surface

10). Not much information or results are provided from the food questionnaire – what were dietary intakes like -what were the most frequently consumed low and high content vitamin D in foods?

11). What does this study mean for policy and food fortification in Japan?

12). This study maybe useful for the authors in some of the discussion/how they display the UV and vitamin D results : 'Ambient UVB dose and sun enjoyment are important predictors of vitamin D status in an older population'. The Journal of nutrition. 2017 May 1;147(5):858-68.

13). Was there any difference with age in terms of vitamin D and UV exposure? Do you see any u-shaped curves from those aged 18-30, 30-60 and >60 yrs in terms of status or UV exposure?

14). Any micro-geographical differences – rural vs urban or suburb vs city?

15). I may have missed this but no information is given on vitamin D supplement intake - what was this in the study and did it differ by demographic criteria

Author Response

Thank you for reviewing our manuscript. Please see the attachment. You are the reviewer #1.

Reviewer 2 Report

In this study, Asakura and colleagues investigate the contributions of dietary vitamin D intake and UV exposure on vitamin D status in people living in two locations in Japan. Their findings suggest that vitamin D intake contributes towards serum 25(OH)D levels throughout the year, while UV exposure in summer (and not winter) further enhances serum 25(OH)D levels. An advantage of this study was that personalised measurements of UV exposure were quantified using a dosimeter, with adjustments made for clothing worn, sunscreen use, skin type and other factors. However, overall, further details are required to improve the quality of this manuscript and clarify various issues throughout. Please find below a list of suggestions from me:

  • Please clarify in the abstract, methods and results whether measurements (25(OH)D, sun exposure, vitamin D intake) were done in the same participants in both summer and winter.
  • Provide the definition of vitamin D deficiency and insufficiency in the abstract.
  • Provide more specific information in the introduction, including: which wavelengths of UV are required for vitamin D synthesis in skin; and, contextualise what seasons each month lies in if reporting specific 25(OH)D levels.
  • The methodology throughout was too brief and did not include sufficient information, including that:
    • It was not clear how participants were recruited. These details should be provided, including means of recruitment, eligibility requirements, numbers of participants who participated at baseline (summer) and then follow-up (winter). A chart would be useful here.
    • Figure 2 really repeats the text above it with a few more details and is not really useful. Could these details be added to the text and Figure 2 deleted?
    • Did the sun exposure, dietary history and physical activity questionnaires use previously validated questions? Please provide references to justify their validation and relative accuracy.
    • Describe how the dosimeter was worn. How compliant were individuals in wearing their badges every day for 10-14 days?
    • It was unclear what the 1T and 2T and 1E and 2E referred to for the statistical modelling. I find these abbreviations confusing.
    • Why was it necessary to adjust for differences in study area? How was this done?
    • It is not clear what the differences are between the two study areas, which should be made clear, including differences in average weather conditions in summer and winter (max/min temp, UV index), other climate/altitude differences, and possible lifestyle differences (e.g. urban v rural).
    • Did you measure the occupation of participants? This should be included in Table 1.
  • Was the LC-MS/MS assay standardised using the vitamin D standardisation program, using NIST 25(OH)D standards? This is important as it is otherwise very difficult to compare rates of vitamin D deficiency/insufficiency across studies, and inaccurate classification can have implications clinically.
  • Why was BMI not measured in both summer and winter? A 6-month gap is a sufficient time period for these to potentially change (and thus influence vitamin D status).
  • Why wasn’t BMI used in the linear regression models?
  • Similarly, physical activity was only measured in winter; however, there could be seasonal differences.
  • Results: There were no statistics done to justify the statements that 25(OH)D levels were greater in men or in the Hokkaido region (paragraph 1).
  • It would be interesting to consider the influence of vitamin D intake and/or UV exposure in maintaining vitamin D status (from summer to winter).
  • I don’t think you need the word ‘quantitatively’ in the title, abstract and throughout when referring to measuring the effects of dietary vitamin D intake and exposure to UV radiation on vitamin D status. It is difficult to know what you mean here. Perhaps you mean personalised measurement using dosimetry?
  • It is interesting that the UV exposure was greater in winter than in summer in the urban area. Are there any predictors of this from your dataset? Please report the finding for the other location in the abstract for comparison.
  • Similarly, it would be interesting to examine factors that modify 25(OH)D according to age (including fish intake, vitamin D intake and UV exposure) as suggested in the discussion.
  • Discussion: I think the first and concluding sentences are likely incorrect. Indeed, there are other studies in which personalised dosimetry has been combined with dietary vitamin D intake to determine the influence of both on vitamin D status (e.g. the AUS-D study. See DOI: 1093/aje/kws322), although perhaps not in Japan.
  • Please elaborate on what you mean by ‘tertiary industries’ on line 202.
  • Some close editing to improve the quality of the English is needed. For example, I don’t know what ‘internal vitamin D status’ refers to in the title. There are many examples throughout the text of awkward phrasing and other errors which should be improved and/or corrected. Vitamin D is not a pronoun.

Author Response

Thank you for reviewing our manuscript. Please see the attachment. The attachment includes our reply to the two reviewers. You are the reviewer #2.

Round 2

Reviewer 2 Report

The authors have largely addressed initial concerns raised by both reviewers. However, there are still a few, which need further clarification.

The authors have not really addressed whether they used a standardised vitamin D assay, as requested previously. This is important for future researchers to be able compare data on rates of vitamin D deficiency within Japan and to other international populations.

I am not sure that I agree with the authors in that they “couldn’t find any evidence to support the direct relationship of weight change with serum vitamin D concentration, vitamin D intake, and UV exposure”.  There is quite a bit of evidence around the associations between reduced serum 25(OH)D and people of higher BMI. For example, see doi: 10.1093/nutrit/nuy034.

Why not include the table that shows differences between age groups in vitamin D intake, UV exposure and fish intake in the paper? This should be slightly modified to show more data detail (mean exposure/intake with a measure of variation, number of participants, etc) and properly formatted, and also include other nutritional intakes relevant to vitamin D (including eggs, mushrooms etc). Were there any statistical differences across the age groups?

I still think the concluding line: “In particular, this study is the first to measure and describe UV exposure time in detail, a variable which cannot be measured with polysulphone film…” is not quite accurate. Perhaps it would be better to state it is the first study to compare the influence of time of exposure, with energy of exposure and serum 25(OH)D levels, with both objectively measured via electronic dosimeters?

Please provide the reasoning for exclusion of pregnant or lactating women.

At line 37, please check that you have the correct wavelengths for UVB (I think should be 280 to 315 nm).

In Section 2.1, please describe in more detail how participants were recruited. E.g. via telephone invitation from the electoral role, or via social media, or leaflets, or other (etc).

It would be better to choose a study in which you report validity of the DHQ questionnaire for non-pregnant women, especially as you have chosen to exclude this population from your study (from line 129). Include p-values for any reported correlations tests detailed in this section.

At line 146, please clarify what you mean by ‘NIST traceable’.

From line 263, I suggest you change ‘20s’ to ‘twenties’ and ‘60s’ to sixties’ (or another abbreviation) as there are a lot of numbers in this sentence already.

From line 299, please also provide details on what program you used to perform the power analysis, for which type of test was used.

Please delete the word ‘internal’ from the phrase ‘internal vitamin D status’ throughout.

Author Response

Thank you so much again for reviewing our manuscript. Please see the attachment.
